# Impacts of Chitosan Coating on Shelf Life and Quality of Ready-to-Cook Beef Seekh Kabab During Refrigeration Storage

**DOI:** 10.3390/foods14223844

**Published:** 2025-11-10

**Authors:** Zubair Hussain, Muawuz Ijaz, Min Li, Kalekristos Yohannes Woldemariam, Zhiying Wang, Dongmin Liu, Chao Wu, Xin Li, Qiankun Zheng, Dequan Zhang

**Affiliations:** 1Key Laboratory of Agro-Products Quality and Safety in Harvest, Storage, Transportation, Management and Control, Institute of Food Science and Technology, Chinese Academy of Agricultural Sciences, Ministry of Agriculture and Rural Affairs, Beijing 100193, China; zubairaja530@gmail.com (Z.H.); dequan_zhang0118@126.com (D.Z.); 2Delisi Technology Center for Postdoctoral Research Work Station, Shandong Dingke Testing Technology Co., Ltd., Delisi Group Co., Ltd., Weifang 262216, China; dlslimin1982@163.com (M.L.); singwoodzxl@126.com (Z.W.); lielie871120@126.com (D.L.); 18732491048@163.com (C.W.); zqk151314@163.com (Q.Z.); 3Department of Animal Sciences, College of Veterinary and Animal Sciences, University of Veterinary and Animal Sciences, Jhang 35200, Pakistan; muawuz.ijaz@uvas.edu.pk; 4National Center of Technology Innovation for Pigs, Food Processing Research Institute, Chongqing Academy of Animal Science, Chongqing 402460, China; 15611560719@163.com

**Keywords:** beef, ready-to-cook, coating, lipid oxidation, antimicrobial activity, storage

## Abstract

Ready-to-cook (RTC) meat products provide convenience but are more susceptible to quality degradation during refrigerated storage. This study examined the effects of 1%, 2%, and 3% chitosan coating on the quality parameters and shelf life of pretreated seekh kabab samples compared with non-coated seekh kabab (NC-SK) samples stored at 4 °C for 28 days. The results show that the chitosan-coated seekh kabab (CC-SK) samples had higher lightness and stable redness values during storage compared with the NC-SK samples. The moisture loss was significantly higher (*p* < 0.05) in the NC-SK samples on days 21 and 28. The chitosan coating effectively retarded lipid oxidation, protein oxidation, and total viable basic nitrogen formation and preserved the natural pH during storage from days 14 to 28 compared with the NC-SK samples (*p* < 0.05). After 21 days, the total viable count, Enterobacteriaceae, LAB, and *S. aureus* counts were higher (*p* < 0.05) in the NC-SK samples than in the CC-SK samples. The sensory scores of the NC-SK samples fell below the acceptable limit compared with those of the CC-SK samples on day 28. A combined effect of refrigeration temperature and chitosan coating enables long storage time, prevents microbial growth, and minimizes lipid oxidation.

## 1. Introduction

Ready-to-cook meat products draw attention from consumers due to their quick and convenient preparation. Kababs are popular ready-to-cook meat products used as appetizers and snacks in India, Pakistan, Bangladesh, Turkey, Iran, Saudi Arabia, and other Middle Eastern countries [1]. According to the latest research, the global kabab shop market size reached USD 21.7 billion in 2024, reflecting a robust food service sector worldwide. The market is experiencing steady expansion, registering a compound annual growth rate (CAGR) of 6.2% from 2025 to 2033 [2]. It requires minimal cooking before consumption [3]. Among different types of kababs, beef seekh kababs attract the greatest attention from customers. Seekh kababs are usually prepared from beef, mutton, chicken, lamb, and fish, marinated with different kinds of spices and condiments. It is popular in the market due to its unique taste, flavor, and shape. The challenges in the production and modification of meat products are standardization of the ingredients, consistency in product composition, and the securing of the product’s safety, quality, and shelf life during storage [4]. Inappropriate storage conditions can lead to lipid oxidation and increase microbial contamination of food products [5], resulting in deterioration of their sensory, nutritional, and physicochemical characteristics [6].

Over the last few decades, the use of natural polymers to support the extension of shelf life in meat products has attracted greater attention in the food industry due to their eco-friendly nature for human health [7]. Among different natural polymers, chitosan stands out for its high applicability in food products. It is one of the most popular biopolymers obtained from the deacetylation of chitin, with a structure of (1–4)-linked 2-amino-2-deoxy-β-d-glucose and 2-acetamido-2 deoxy β-d glucose. It contains two functional groups, including amino (-NH_2_) and hydroxyl (-OH) [8]. It is generally obtained from living organisms including insects, crustaceans, and fungi, which are biodegradable, non-toxic, and biocompatible [9]. According to the Food and Drug Administration (FDA), it is generally recognized as safe (GRAS) for addition in food products as an additive [10].

Chitosan possesses numerous functional properties which make it a valuable material for food preservation. It is widely used as an antimicrobial agent and antioxidant in fruit, vegetable, and meat products [11]. Additionally, its film forming ability and gas barrier properties further enhance its suitability for preserving meat products [12]. It can be used in many ways: direct addition during new-product formulation, as edible coating, or for making packaging films to improve the quality, safety, and shelf life of meat products during storage [13]. Additionally, chitosan coating has been shown to be effective in increasing the quality of meat products through a number of mechanisms, including oxygen diffusion reduction, its function of carrier for food additives, product appearance improvement, the delay of moisture loss, and the delay of lipid oxidation and discoloration [14].

Chitosan has been widely utilized as a bio-preservative in various meat products to enhance quality, extend shelf life, and ensure safety [7]. Some earlier studies have explored the application of chitosan as a non-meat protein extender, as well as in meat products such as chevon and chicken seekh kabab, with varying formulation, processing methods, and storage conditions [3,15]. For instance, [16] demonstrated that chitosan-coated chicken breast enables better prevention against pathogenic microbial growth, extending shelf life. However, gaps remain in understanding how different levels of chitosan coating influence the time-dependent changes in quality parameters such as lipid oxidation, microbial stability, and sensory attributes particularly in ready-to-cook seekh kabab, which is increasingly popular across several countries.

The selection of concentrations was based on previous studies [17,18] and preliminary tests. It was observed that chitosan concentrations above 3% led to undesirable changes in the surface appearance and altered taste (slightly bitter) of the meat, potentially affecting consumer acceptability. Conversely, coating concentrations below 1% exhibited limited antimicrobial efficacy. Therefore, chitosan concentrations ranging from 1% to 3% were selected as the treatment levels in this study to determine the optimal concentration for enhancing the microbial stability, quality, and shelf life of coated meat samples.

## 2. Materials and Methods

### 2.1. Materials

Low-molecular-weight chitosan powder (food grade; CAS No. 70694-72-3; molecular weight of 50–190 kDa; Batch No. 230223) sourced from crustacean shells with a deacetylation degree of 95%, a purity of 87.7%, and a moisture below 10% (dry basis) was purchased from Qingdao Honghai Bio-Tec Co., Ltd. Qingdao, China. Glycerol was obtained from Tianjin Haihua Trading Chemicals Co., Ltd. Tianjin, China. Polyethylene glycol was received from Shandong Ruisheng Pharmaceutical Ingredients Co., Ltd., Weifang, China.

### 2.2. Raw Material Collection

Beef meat (*longissimus thoracis et lumborum* (LTL)) and fat were collected from a local market 48 h postmortem, placed in an icebox after having been packed in a zip locker bag, and transferred to the lab within 2 h of collection. The animals were of the same sex, batch, and breed and had been subjected to the same pre-slaughter treatments and feeding system. They ranged in age from 18 to 24 months. The Halal slaughtering method, which included exsanguination without electrical stunning, was followed to kill all the animals on the same day. The average weight of the carcasses, which were not electrically stimulated, was 351 kg (standard error: 11.19 kg). All the other ingredients used during the preparation of seekh kabab, which were purchased from the local market, are presented in Table 1.

### 2.3. Chitosan Coating Preparation

The formulation of the chitosan coating mixture was prepared by following the protocols described by [3]. In detail, chitosan at different concentrations, 0%, 1%, 2%, and 3%, was dissolved in distilled water using a stirrer (INTLLAB Magnetic Stirrer Mixer 3000 rpm, INTLLAB, Foshan, China) for 10 min at 60 °C. After that, 1.5% of plasticizer (consisting of 50% polyethylene glycol and 50% glycerol) was added with 1% of acetic acid to the chitosan solution. The use of the plasticizer was intended to optimize the mechanical and functional properties of the chitosan coating, whereas 1% acetic acid was used to dissolve the chitosan and ensure uniform dispersion of the plasticizer. The prepared solution was placed in a hot plate magnetic stirrer (RCT-B-S25, IKA, Königswinter, Germany) for proper mixing at 1400 rpm for approximately 30 min at 60 °C.

### 2.4. Seekh Kabab Preparation

The collected meat and fat were ground twice using a 6 mm grinding plate in a meat mincing machine (Duronic Meat Grinder MG301, Duronic International Ltd., Romford, UK). The quantity and acceptability of all ingredients used in the seekh kabab preparation were ensured according to Table 1. The non-meat ingredients, including onion, green chili, garlic, and ginger, were ground for 2 min using a grinder (FP 3010 Food processor, BRAUA, Frankfurt, Germany). The onion was squeezed using a mesh cloth after grinding to remove excess water content. The resulting paste of non-meat ingredients was added to the minced meat and mixed manually for 3 min. Definite amounts of spices (Table 1), i.e., cumin, black pepper, red pepper, long pepper, and coriander, were ground using an electric herb grinder (Model-750, Lejieyin, Ningbo, China). The spice powders, lime powder, and baking soda were then mixed into the meat mixture and placed into the refrigerator at 4 °C for 1 h to enhance marination. After marination, the blended meat was molded onto pre-sterilized iron skewers. Each beef seekh kabab weighed approximately 50 g, consisting of 70% meat (80% meat and 20% fat) and 30% other ingredients. The molded seekh kabab was then precooked for 3 to 4 min on an electric barbecue grill. After that, the seekh kabab was removed from the griller machine and cooled to room temperature for 30 min.

Four different batches of seekh kababs were produced to evaluate the effects of chitosan coating. One batch was considered a control without chitosan coating (NC-SK), while the other three batches were treated with varying levels of chitosan coating. The batches were labeled as follows: control (NC-SK), 1%, 2%, and 3% chitosan coating. The precooked seekh kabab samples were dipped in a chitosan solution (1%, 2%, and 3%) for 2 min to ensure the complete coating. The samples were then drained and allowed to fully dry under laminar airflow (HH 48, Holten LaminAir, Thermo Fisher Scientific, Bath, UK) at an air velocity of 0.35–0.45 ms^−1^ for 30 min. Following this, the samples were properly labeled and then packaged in commercial low-density polyethylene (LDPE) plastic zip locker bags (thickness of 40–50 µm, O_2_ permeability, and 80 mL O_2_/m^2^/day^−1^) and stored at 4 °C for intervals of 0, 7, 14, 21, and 28 days. At the end of every storage period, the samples were taken for biochemical examination. A total of 320 seekh kababs were produced, with 8 seekh kababs for each treatment at each storage time per replication (4 for biochemical analyses and 4 for sensory evaluations). The experiment was conducted with two biological replications, meaning that the entire process (formulation, coating, drying, packaging, and storage) was independently repeated on two separate occasions by using fresh raw materials. To control variability, all processing steps were standardized, and the same equipment and protocols were used across both replicates.

### 2.5. Evaluation of Antimicrobial Activity of Chitosan Coating

The disc diffusion method was used to assess the anti-pathogenicity of the chitosan coating solution against two common foodborne pathogens, *Escherichia coli* (*E. coli*, CISS-10389) and *Staphylococcus aureus* (*S. aureus,* CICC-10201), in accordance with the methodology of [19]. Initially, 100 µL of *E. coli* (1 × 10^8^ CFU/mL) and *S. aureus* (1 × 10^8^ CFU/mL) bacteria was spread over agar plates, and a pre-sterilized filter paper disc (6 mm) was placed on the inoculated surface. Subsequently, 10 µL of chitosan solution was applied onto to the disc and incubated at 37 °C for 24 h. Following incubation, antibacterial activity was determined by measuring the diameter of the inhibition zone with a scale.

### 2.6. Analysis of Seekh Kabab Samples

#### 2.6.1. Determination of Proximate Analysis

The chemical composition in the RTC seekh kabab samples including moisture, fat, protein, and ash contents was measured according to the AOAC official methods (950.46, 991.36, 928.08, and 920.153), respectively.

#### 2.6.2. pH

pH was assessed by following the procedure described by [20]. A pH meter (Testo 205 pH meter, Lenzkirch, Germany) was directly injected into four different locations of seekh kabab at the end of each storage period.

#### 2.6.3. Color

The color values of L* (lightness), a* (redness), and b* (yellowness) of seekh kabab during each storage interval were measured using a calorimeter (CR-400, Konica Minolta Sensing Inc., Osaka, Japan) [21]. The device was standardized using a Minolta calibration plate before recording the color values in different locations of samples at the end of each storage period.

#### 2.6.4. Lipid Oxidation

The thiobarbituric acid reactive substances (TBARS) method was used to assess lipid oxidation in all seekh kabab samples following the method described by [22]. Ten grams of seekh kabab samples was measured in a test tube containing 25 mL of 25% (*w*/*v*) trichloroacetic acid (TCA) and blended for 30 s using a homogenizer (SCI-ENTZ-11, Ningbo Scientz Biotechnology Co., Ltd., Ningbo, China). After that, the blended samples were centrifuged (TG16-WS Benchtop High-Speed Centrifuge, Changsha, China) at 4000 rpm for 30 min at 10 °C. Then, the supernatant from each sample was collected using Whatman filter paper, and 2 mL was transferred into Eppendorf tubes that contained 2 mL of thiobarbituric acid (20 mM). The samples were then vortexed (QL-861, Ningbo Hinotek Technology Co., Ltd., Ningbo, China) for 5 s and heated in a thermostatic bath at 97 °C for 20 min. The heated samples were then cooled under running water and absorbance was taken at 532 nm using a spectrophotometer (UV 6100, Shanghai Metash Instruments Co., Ltd., Shanghai, China). Finally, the TBARS values mainly malondialdehyde (MDA) values were calculated using a standard curve for 1,1,3,3 tetra-ethoxypropane in an equimolar reaction.

#### 2.6.5. Protein Oxidation

Carbonyls in proteins were measured using the methodology outlined by [23]. Utilizing the 2,4-dinitrophenylhydrazine (DNPH) technique, the carbonyl content was determined. A 1 g sample was taken from the lamb patties and homogenized for 30 s in 10 mL of 20 mM phosphate buffer containing 0.6 M NaCl. Two 2 mL aliquots of the homogenate (designated S1 and S2) were transferred into separate 2 mL Eppendorf tubes. Then, 1 mL of 10% TCA was added to each tube to precipitate the proteins, followed by centrifugation at 5000 rpm for 5 min. After centrifugation, S1 was treated with 1 mL of 0.2% DNPH in 2 M HCl, while S2 (the control) received 1 mL of 2 M HCl alone. Both samples were then incubated at room temperature for 1 h. Subsequently, 1 mL of 10% TCA was added again to precipitate the samples, followed by another centrifugation step (5000 rpm, 5 min), after which the supernatant was discarded. To remove excess DNPH, the pellets were washed with 1 mL of an ethanol–ethyl acetate mixture (1:1, *v*/*v*). The pellets were then dissolved in 1.5 mL of 20 mM sodium phosphate buffer containing 6 M guanidine HCl (pH 6.5) and centrifuged at 5000 rpm for 2 min to remove insoluble debris. Finally, the absorbance of the supernatant was measured. Protein concentration was measured at 280 nm using BSA (bovine serum albumin) as a reference. With an absorbance value of 21.0 nM^−1^ cm^−1^ at 370 nm for protein hydrazones, the carbonyl concentration was represented as nmol of carbonyl per mg of protein.

#### 2.6.6. Total Volatile Basic Nitrogen (TVB-N)

TVB-N analysis was performed as explained by Chinese standard method GB/T 5009.44 (2003). Briefly, 10 g of minced seekh kabab sample was homogenized with 75 mL of distilled water and shaken for 10 min. After that, 1 g of magnesium oxide (MgO) was properly mixed with the mixture containing water and the seekh kabab sample. A KDN-103F Kjeldahl apparatus (Shanghai Qianjian Instrument Co., Ltd., Shanghai, China) was then used to distill the blended samples. The distillate was absorbed into a 25 g/L H3BO4 solution containing mixed indicators of methyl red and bromocresol green and then titrated with 0.01 M HCl. The results were expressed as mg of TVB-N per 100 g of meat.

#### 2.6.7. Microbiological Analysis

Twenty-five grams of seekh kabab samples were aseptically weighed and homogenized with 225 mL of 0.1% saline water in a stomacher bag using a stomacher (TF-08, TEFIC Biotech Co., Ltd. Xian, China) for 2 min at room temperature [24]. Serial decimal dilutions were prepared in 0.1% saline water for each sample; a volume of 0.1 mL of sample dilution was transferred into the selected agar plates, and the samples were dispersed on the agars following a spread method. Following 48 h of incubation at 37 °C, total viable counts (TVCs) were performed in Plate Count Agar (PCA; Oxoid, Unipath Ltd., Basingstoke, UK). However, following three days of incubation at 37 °C, lactic acid bacteria were identified on de Man–Rogosa–Sharpe (MRS) medium agar (Oxoid, Unipath Ltd., Basingstoke, UK). Enterobacteriaceae were counted on Violet Red Bile Glucose (VRBG; Oxoid, Unipath Ltd., Basingstoke, UK) plates after incubation at 37 °C for 24 h. Staphylococcus aureus was counted using Mannitol Salt Agar (MSA; Oxoid, Unipath Ltd., Basingstoke, UK). Petri dishes with 30–300 colonies were considered for result interpretation after a specific number of days of incubation time. The results were expressed as logarithms of the number of colony-forming units (CFU/g).

### 2.7. Sensory Properties of Cooked Seekh Kabab

A total of 30 panelists (representing the staff of Delisi Group Co., Ltd., Weifang, China) were trained on how to evaluate and also briefed about the product type and composition [22]. Among the selected 30 panelists, 21 were selected through pilot-phase discrimination analysis scoring. A total of 21 trained panelists participated in evaluating the sensory attributes of the RTC seekh kabab samples using a 7-point numerical scale [25]. The judges included 8 males and 13 females in the age group of 25 to 45 years. Sensory analysis was conducted in accordance with the Declaration of Helsinki and was approved by the Ethics Committee of the Institute of Food Science and Technology, Chinese Academy of Agricultural Sciences (CAAS-IFST and DLS-2025/267 on 5 April 2025). The sensory qualities of seekh kabab (texture, flavor, taste, juiciness, and overall acceptability) were rated by the staff members who participated in the sensory evaluation. In order to prevent bias, seekh kabab samples were given to panelists in disposable white plates at a sensory evaluation booth after being randomly tagged with a three-digit number. For each sampling period, three sensory sessions were conducted, and each panelist received four samples. Each panelist was provided with water and plain crackers to cleanse the palate between samples. Samples were presented in a randomized order to avoid order bias. The scores ranged from “extremely like” = 7 to “extremely dislike” = 1, and 4 was taken as the lower limit of acceptability; the tests were replicated at each storage interval. The mean scores from all judges were calculated and analyzed for the sample and the session.

### 2.8. Statistical Analysis

The statistical analysis was performed using SPSS 21 software (SPSS Inc., Chicago, IL, USA). The results were expressed as means ± SDs. The effects of key factors such as different levels of chitosan coating and storage periods on various dependent variables, such as ash, moisture, fat, protein, pH, L*, a*, b*, TBARS, TVB-N, TVC, LAB, *S. aureus*, Enterobacteriaceae, and sensory attributes, were analyzed using General Linear Model (GLM) multivariate analysis. The level of significance (*p* < 0.05) was evaluated using Duncan’s multiple range test. Correlation analysis was performed using Origin Pro student version, (OriginPro, 2023, OriginLab Corporation, Northampton, MA, USA) using a scatter matrix plot at a 95% confidence interval using Pearson (r) and Adj. Pearson (R^2^).

## 3. Results and Discussion

### 3.1. Antimicrobial Activity of Chitosan Coating

Antibacterial packaging has become a promising technique to prevent food contamination and the growth of pathogenic microorganisms [26]. The efficacy of antimicrobial packaging coating was determined by observing the diameter of the inhibition zone. The antibacterial activity of 1% CC-SK was significantly (*p* < 0.05) lower against *S. aureus* compared with other test coating concentrations, as depicted in Figure 1. However, there was no significant difference (*p* < 0.05) in *S. aureus* between 2% CC-SK and 3% CC-SK. In contrast, chitosan coating was more effective in inhibiting the growth of Gram-negative bacteria.

All coating concentrations, with 1% CC-SK > 2% CC-SK > 3% CC-SK, reduced the growth of Gram-negative bacteria. The inhibition zone for *S. aureus* ranged from 9.13 ± 0.23 to 12.34 ± 0.35 mm, while that for *E. coli* ranged from 11.67 ± 0.31 to 15.23 ± 0.27 mm, respectively indicating effective inhibition against both Gram-positive and Gram-negative bacteria. Notably, 3% CC-SK exhibited a larger inhibition zone (15.23 ± 0.27 mm) against Gram-negative bacteria compared with that (12.34 ± 0.35 mm) against Gram-positive bacteria, possibly due to the higher sensitivity of Gram-negative bacteria to chitosan derivatives. These findings are aligned with those of [19], who applied carboxymethyl chitosan coating solution at concentrations of 2% and 4% to mango fruit and reported significant shelf-life extension, supporting the suitability of these coating levels against Gram-positive (*S. aureus)* and Gram-negative (*E. coli*) bacterial strains for food preservation applications. Chitosan functions as an antibacterial agent through electrostatic interactions between its positively charged amino groups and the negatively charged microbial cell membranes, increasing membrane permeability, causing leakage into intracellular contents, and ultimately resulting in cell death [12,27].

### 3.2. Proximate Analysis

The effects of varying levels of chitosan coating on the ash, moisture, fat, and protein of the seekh kabab samples over a storage period of 28 days at 4 °C are demonstrated in Table 2. The ash content in the NC-SK and CC-SK samples on day 0 was significantly higher (*p* < 0.05) compared with day 28. However, among all CC-SK and NC-SK treatments, the ash content in the 2% CC-SK treatment samples showed significantly lower values from day 7 to day 28 compared with NC-SK samples during the same storage periods. The moisture content in the NC-SK samples was significantly higher (*p* < 0.05) at the initial storage time compared with day 21 and day 28 of storage. Similarly, the moisture content in the CC-SK samples was significantly higher (*p* < 0.05) on the initial day and declined after day 14 until the end of storage. In addition, the moisture content in all CC-SK samples was higher on days 21 and 28 compared with the NC-SK samples. The difference in the composition of moisture and protein contents in chitosan nanoparticle-coated chicken breast meat might be due to coating concentrations [28]. Among other related studies, Ref. [3] noted similar results in the loss of moisture content in chevon kabab meat product during storage. The highest moisture content was noted in the CC-SK samples during storage, which might be due to the water retention property of chitosan. It has been reported that chitosan forms semipermeable coating that can decrease water migration and retard moisture loss from food products [16]. Loss of moisture previously brought unacceptable changes in texture, flavor, color, and saleable weight of meat and meat products during storage [7].

The fat content showed a decreasing trend and exhibited significantly higher values on day 0 and day 7 for all CC-SK and NC-SK samples during storage. The fat content in the NC-SK samples showed significantly lower values on day 21 and day 28 compared with the coated samples. Additionally, it was found that the protein content in all samples exhibited higher values on day 28 compared with day 0 of refrigeration storage. Moreover, crude protein was significantly higher in the 2% and 3% CC-SK samples from day 21 to day 28 than in the NC-SK samples. The decline in protein content in uncoated meat samples is likely due to proteolytic enzyme activity and microbial spoilage, which degrade protein into peptides and free amino acids. However, chitosan coating acts as an antimicrobial agent and has film-forming properties, thereby slowing down the microbial spoilage process and inhibiting oxidative reactions [29].

### 3.3. Change in pH of Seekh Kabab

The difference in the pH values of all RTC seekh kabab samples with various levels of chitosan coating during refrigerated storage is presented in Table 3. The initial (day 0) pH for all NC-SK and CC-SK samples was in the range of 5.72 to 5.80, and significantly higher values were obtained at the end of the storage period. In addition, between the NC-SK and CC-SK samples, the pH value of the NC-SK samples was significantly higher (*p* < 0.05) on day 14 to day 28 than that of all CC-SK samples. The findings of the current study regarding pH agree with those of [3], who observed higher pH values in non-chitosan seekh kabab samples than in chitosan-treated samples on the 30th day of storage. However, it has been shown that chitosan coating can sustain the natural pH of meat products [30]. The higher pH values in the NC-SK samples might be due to microbial and enzymatic activities, which lead to the accumulation of volatile metabolites like ammonia, amines, free amino acids, and other alkalines, as well as degradation of proteins [31].

### 3.4. Chromaticity and Color Determination of Seekh Kabab

Color is a quality parameter used to assess the freshness of meat products, and consumers often consider it when purchasing meat products. The changes in the color values of lightness (L*), redness (a*), and brownness (b*) of the ready-to-cook NC-SK and CC-SK samples during refrigeration storage are presented in Table 3. In the present study, the initial instrumental color values of L* were in the range from 43.08 ± 0.75 to 44.12 ± 1.03; those of a*, from 7.89 ± 0.10 to 8.05 ± 0.27; and those of b*, from 18.07 ± 1.14 to 19.04 ± 0.76. These results are in line with previous studies [3], which reported a 0-day L* value range of 43.19 to 44.09; similarly the a* values were in the range of 9.33 to 10.61 and the b* values in the range from 21.35 to 22.96 in chitosan-coated chevon kabab samples. In another study, ref. [32] stated that the L* value of an uncoated sample was 43.34 in beef patties with a slight increase after chitosan coating. Regarding the storage periods, the L* values of the 1% and 2% CC-SK samples and NC-SK samples were higher (*p* < 0.05) from day 21 to day 28 during storage, unlike those of the 3% CC-SK samples. However, the higher the level of chitosan coating of seekh kabab samples, the higher the L* value; the L* values were significantly higher (*p* < 0.05) for the 2% and 3% CC-SK samples compared with the NC-SK samples from day 14 to day 28 during storage. Previous studies confirmed that higher L* values in ready-to-cook samples indicate the presence of chitosan coating compared with control samples [17].

For all treatment samples, there were no significant changes in the a* values throughout the first two storage periods. However, on day 21 and day 28, the a* values of the 1% CC-SK samples and NC-SK samples showed significantly lower values than on other days (*p* < 0.05). Pigment oxidation is frequently blamed for a reduction in redness, as seen by a lower a* value [33]. Myoglobin, an oxygen-carrying protein, is the main cause of meat’s red hue. The oxidation of myoglobin results in the formation of brown metmyoglobin, which lowers redness (a* value) during storage [34]. Regarding the treatments, the a* value of 2% and 3% CC-SK was significantly higher (*p* < 0.05) on day 21 and day 28 compared with the NC-SK samples. Ref. [35] observed an improvement in the L* and a* values of chitosan-treated pork fillets during storage compared with control samples. Another study found that chitosan coating reduced TBARS and metmyoglobin formation, thereby retaining L* and a* values during storage [36]. The b* values of the NC-SK and CC-SK samples increased with storage time. The b* values of the 1% and 2% CC-SK samples and NC-SK samples were significantly higher (*p* < 0.05) from day 21 to day 28 than on day 0 and day 7 of storage. However, among the treatments, the b* value of the 2% CC-SK samples was significantly lower than that of the NC-SK samples during storage.

It has been stated that the decline in a* value may result from the partial loss of red color in meat caused by the oxidation of oxymyoglobin (Fe^2+^) to metmyoglobin (Fe^3+^) [37]. Additionally, the oxidation of fat has been linked to an increase in b* values [7]. Studies have continuously reported that chitosan has the potential to improve the color of meat products by limiting unfavorable changes in meat color during storage [38]. Interestingly, in the current study, we noticed that samples with less lipid oxidation were color-stable; for instance, the 2% and 3% CC-SK samples were color-stable and presented less lipid oxidation. Earlier studies confirmed that the color properties of meat samples during storage is greatly influenced by lipid oxidation [39]. The pigment that gives meat its color, myoglobin, can become unstable and discolored due to increased lipid oxidation. Furthermore, ref. [40] concluded that the color of meat deteriorates during storage due to heme-protein oxidation, which also causes rancidity, which has a significant impact on customer acceptance. A connection between lipid and heme-protein oxidation was discovered by many authors. Actually, the oxidation of lipids and heme-proteins in meat happens simultaneously, and one step seems to benefit the other [37].

### 3.5. Secondary Lipid Oxidation in Seekh Kabab

Lipid oxidation is one of the important parameters in determining the quality of meat products during storage. The TBARS values indicate the amount of lipid oxidation of secondary products, which is generally used to evaluate the degree of lipid oxidation of meat products. The lipid oxidation in the NC-SK and CC-SK samples was obtained using TBARS values in mg MDA/kg, which are shown in Table 4. It was noted that all seekh kabab samples showed higher TBARS values ranging from 1.57 to 1.64 mg MDA/kg on the initial day (*p* < 0.05). In line with the current findings, previous studies have documented TBARS values of cooked meat on day 0 of 1.55 to 1.78 mg MDA/kg and 1.17 to 1.98 mg MDA/kg [41]. Ref. [42] reported that high temperature and marinade composition influenced the TBARS values of the meat product. In general, the TBARS values of all samples significantly increased on day 7 compared with day 0. The TBARS values of the NC-SK samples were significantly higher (*p* < 0.05) compared with those of the CC-SK treatment samples on days 14, 21, and 28, with 2.32, 2.57, and 3.20 mg MDA/kg, respectively. On day 28, the 1%, 2%, and 3% CC-SK samples significantly retarded the formation of TBARS compared with NC-SK (*p* < 0.05). The comparatively slow increase in the TBARS values of the CC-SK samples might be due to chitosan antioxidant activity.

The higher lipid oxidation in the NC-SK stored samples might be due to the direct contact of oxygen with the samples, whereas the treated samples were covered with the coating. Previous studies reported that chitosan coating retarded lipid oxidation (MDA) to 0.47 mg/kg after 14 days compared with uncoated samples in refrigerated storage [43]. Chitosan slowed down lipid oxidation to 1.18 mg/kg, while the TBARS values of uncoated meat samples reached 1.66 mg/kg, exceeding sensory thresholds [44]. Applying chitosan coating on RTC pork chops slowed down the lipid oxidation process in the coated samples (1.02 ± 0.03 MDA mg/kg) after 12 days compared with uncoated samples, which reached 2.03 ± 0.04 MDA mg/kg [17]. It has been proven that chitosan coating exhibits oxygen barrier capability and antioxidant properties and helps to retard lipid oxidation in beef products [45]. The antioxidant activity of chitosan is mainly due to the presence of free amino (-NH_2_) and hydroxyl (-OH), which act as hydrogen donors to neutralize lipid radicals and interrupt oxidative chain reactions. Moreover, chitosan can chelate prooxidant metal ions such as Fe^2+^ and Cu^2+,^ thereby inhibiting the Fenton reaction, which generates reactive oxygen species [46]. By chelating transition metal ions or by interacting with malondialdehyde through its main amino group, chitosan can prevent lipid oxidation in meat products [17].

### 3.6. Protein Oxidation of Seekh Kabab

Determining protein carbonyl content is an important technique used to measure oxidative changes in meat samples, which lead to quality degradation and impacts on muscle proteins. The variations in carbonyl content in the coated and non-coated seekh kabab samples are presented in Table 4. The increase in carbonyl content is an indication of the formation of carbonyl compounds, which increased gradually in all treatment groups during the 28-day storage period. The highest carbonyl content was observed in the NC-SK samples from day 7 to day 28, reaching 3.92 ± 0.13 nmol/mg protein by day 28 (*p* < 0.05). The findings are in line with earlier studies highlighting that thermal processing boosts oxidative degradation of proteins in meat products [47]. The application of 2% and 3% chitosan coating significantly (*p* < 0.05) reduced carbonyl formation in seekh kabab samples from day 7 to day 28 compared with the non-coated (NC-SK) samples. On day 28, the 2% and 3% chitosan-coated (CC-SK) samples demonstrated significantly lower carbonyl content (3.42 ± 0.13 and 3.34 ± 0.13 nmol/mg protein, respectively) than the control group (3.92 ± 0.13 nmol/mg protein) (*p* < 0.05). The effectiveness of 2% and 3% chitosan concentrations against protein oxidation might be due to the viscosity of the solution, which forms a thicker coating layer. Similar, ref. [44] reported that increasing chitosan beyond an optimal level did not significantly influence preservation because of limited oxygen permeability and reduced bioactive compound migration. The findings of the current study indicate that the antioxidant activity of chitosan coating prevents carbonyl formation in meat samples. According to [48], higher protein oxidation indicates oxidative stress in muscle proteins, resulting in oxidative degradation of specific amino acid side chains, including proline, lysine, arginine, and histidine residues. However, chitosan coating acts as a protective barrier, limiting oxygen exchange and exhibiting antioxidant properties, thereby slowing down oxidation reactions [8].

### 3.7. Total Viable Basic Nitrogen Composition of Seekh Kabab

TVB-N is among the indicators of food spoilage which can be produced after the degradation of nitrogenous substances in food products. TVB-N showed a clear increasing trend in all NC-SK and CC-SK samples, as illustrated in Table 4. There were no significant differences on the initial day of storage. From day 7 to day 28, the TVB-N values of the NC-SK samples were significantly higher (*p* < 0.05) compared with the initial day of storage. All treatments, including 1%, 2%, and 3% CC-SK, showed significantly higher TVB-N values on day 21 and day 28 compared with the other storage periods. On the 28th day of storage, the 1%, 2%, and 3% CC-SK samples showed reduced TVB-N values, i.e., 2.71, 3.72, and 3.82 mg/100 g, respectively, compared with NC-SK. In the current study, TVB-N was significantly higher (*p* < 0.05) in the NC-SK samples than in the other treatments after 14 days of storage. This aligns with the result of [49], which reported that beef steak samples coated with chitosan and stored for 15, 30, and 45 days could reduce TVB-N contents, which were 6.64, 7.48, and 8.31, respectively. Another study reported that the TVB-N value of chitosan-coated roasted duck groups was lower than that of the control group for 7 days [50]. The breakdown of protein and other nitrogen-containing substances in meat by spoilage bacteria is the cause of the increase in TVBN levels in the NC-SK samples [51].

### 3.8. Microbial Composition of Seekh Kabab

The variations in the proliferation of TVCs, LAB, *S. aureus*, and *Enterobacteriaceae* in the 1%, 2%, and 3% CC-SK and NC-SK samples during storage at 4 °C are presented in Figure 2. The TVCs in seekh kabab during storage at 4 °C were influenced (*p* < 0.05) by the various levels of chitosan coating, as presented in Figure 2a. On the initial day of storage, the TVCs for all CC-SK and NC-SK samples were in the range of 2.69 to 2.81 log CFU/g. On day 14 to day 28 of storage, the NC-SK samples showed significantly higher (*p* < 0.05) TVCs than all CC-SK treatments and reached 7.13 log CFU/g on day 28. The TVCs were reduced on day 28 in the 1%, 2%, and 3% CC-SK samples compared with NC-SK, with values of 0.76, 1.05, and 1.24 log CFU/g, respectively. The influence of various levels of chitosan coating (1%, 2%, and 3%) on the LAB counts in seekh kabab during refrigeration storage is illustrated in Figure 2b. Initially (day 0), no counts were detected for all CC-SK and NC-SK samples during storage. However, the LAB counts in the NC-SK samples were higher starting from day 7 (4.37 log CFU/g) of storage and reached 6.93 log CFU/g on day 28. On day 28, the LAB counts for CC-SK treatment samples were notably lower (*p* < 0.05) than those in the NC-SK samples, while the 3% CC-SK samples showed the lowest counts, i.e., 5.68 log CFU/g, during storage.

The *Enterobacteriaceae* counts in the CC-SK and NC-SK samples during refrigeration storage are presented in Figure 2c. On day 0, no *Enterobacteriaceae* counts were detected for all CC-SK and NC-SK treatment samples during storage. The *Enterobacteriaceae* counts in the NC-SK samples showed a clear upward trend from day 7 to day 28, with significant counts on day 21 (5.72 log CFU/g) and day 28 (6.93 log CFU/g) compared with the CC-SK samples during storage (*p* < 0.05). On day 21, the 3% CC-SK samples exhibited significantly lower *Enterobacteriaceae* counts than all other samples. The counts of *S. aureus* showed an increasing trend for the NC-SK and CC-SK samples, as presented in Figure 2d. *S. aureus* was not detected on day 0; however, from day 7 to day 28, the *S. aureus* counts were significantly higher in NC-SK than in all CC-SK treatment samples, and the highest value reached 5.89 log CFU/g. Significant differences (*p* < 0.05) were also found among CC-SK treatments: the greater the percentage of chitosan coating, the lower the *S. aureus* counts in the samples. The 3% CC-SK samples showed significantly lower (*p* < 0.05) values, 4.48 log CFU/g and 5.78 log CFU/g, on day 21 and day 28, respectively.

The findings of the current study align with those obtained by [18], who stated that 2% chitosan coating retarded the proliferation of LAB and aerobic bacteria in red sausage compared with the control samples. Moreover, ref. [52] reported that dipping sucuk (Turkish dry fermented sausage) in chitosan solution retarded the proliferation of Enterobacteriaceae during storage. Ref. [49] found that LAB, *S. aureus*, and total mesophilic aerobic bacteria counts were observed to be lower in chitosan-treated beef loin compared with the control. The inhibitory effect on the CC-SK samples could be due to the antioxidant and antimicrobial properties of chitosan [7]. Numerous studies have reported the correlation between shelf-life extension and antimicrobial activities of chitosan coating in meat products during storage [53]. Chitosan’s antibacterial properties have been attributed to its cationic property, which permits electrostatic interaction between the negative charges of microbial cell membranes and the positive charge on the NH_3_ group of glucosamine monomer in chitosan molecules. This interaction causes intracellular constituents to leak out and influences protein expression [12,54]. Moreover, the selective permeability of chitosan reduces the entrance of oxygen into the meat may constitute unfavorable conditions for microbial growth [55].

### 3.9. Sensory Attributes

The sensory scores for color, odor, texture, flavor, and overall acceptability of the NC-SK and CC-SK treatment samples during refrigerated storage for 28 days are presented in Figure 3. The sensory scores of all treatment samples were significantly lower on day 21 and day 28 compared with day 0 and day 7 of storage (*p* < 0.05). The sensory scores for quality parameters of 2% CC-SK, including color, odor, texture, flavor, and overall acceptability, were significantly higher (*p* < 0.05) from day 14 to day 28 than those of the NC-SK samples during storage, except for texture on day 14. On the 28th day of storage, the NC-SK samples were rejected by the judges and received lower sensory scores (lower than 4 points) than the acceptable limit. A score of 4 was taken as the lower limit of acceptability (neither like nor dislike) [25]. Overall, the CC-SK samples showed higher sensory scores throughout storage compared with the control samples. The results are in line with earlier studies which reported that RTC pork chops coated with chitosan maintained the sensory attributes and could extend shelf life during storage compared with the control [33]. In previous studies, it was illustrated that applying chitosan coating to RTC chicken products showed more acceptable odor and taste compared with the control group [56]. According to previous studies reported by [25], RTE meat products with chitosan had higher overall acceptance ratings than samples without coating. The higher sensory score of CC-SK samples during storage might be due to microbiological stability and retardation of oxidation due to chitosan. Furthermore, the lower sensory scores of the NC-SK samples might be due to the loss of moisture during storage. It was reported that the use of polyethylene freezer bags may have influenced volatile compound retention, moisture, and sensory attributes [57]. It was stated that the loss of moisture brought unacceptable changes in texture, flavor, color, and saleable weight of meat and meat products during storage [7].

### 3.10. Correlation Analyses on Effect of Storage Time

Based on the significance and importance of SK shelf life, correlation analyses for TVB-N, TBARS, pH, and moisture were performed to study the effect of storage time on other factors, as shown in Figure 4. The highest correlations with TVB-N and TBARS were recorded at 28 days, with r and R^2^ of 0.86 and 0.71, with these factors contributing to an increase in the storage time of SK samples. This shows the similarity in the increase in TVB-N and TBARS in samples with the increase in time. The increase in TVB-N and TBARS during storage is mainly associated with proteolysis and microbial action in SK samples. TVB-N is an indicator of the hydrolysis of proteins and an increase in volatile nitrogen, which can be associated with the presence of bacterial metabolism [58]. The increase in TBARS is an indication of secondary lipid oxidation, which can be initiated as a result of endogenous lipid oxidation enzymes or bacterial lipid oxidation. Considering these factors, the levels of TVB-N and TBARS align with the minimum concentration in the 2% and 3% chitosan-coated samples, indicating the high preservative effect of chitosan through the prevention of bacterial growth and the antioxidant effect of low-molecular-weight chitosan [59].

The correlation between pH and moisture shows less correlation with the total Pearson r and R^2^ of 0.71 and 0.48, which indicates that the change in moisture does not influence the change in pH of SK. During storage, the moisture content in all SK samples is high at 0 d and decreases with storage time, while the change in pH shows an increase with storage time. The inverse relation between pH and moisture mainly relates to chitosan, as it prevents the growth of bacteria and minimizes LAB growth, while protein hydrolysis and the formation of TVB-N contribute to an increase in free amine groups, resulting in an increase in pH. The decrease in moisture mainly relates to the permeability of chitosan coating. As a study indicates, it allows for a better preservative effect with minimal coating, attaining optimum moisture content during the storage of chicken meat [60]. A study also shows that the increase in TVBN and TBARS is mainly a result of endogenous and microbial enzymatic activity on proteins and lipids during storage and is retarded by chitosan coating [3].

The present study showed that chitosan coating successfully enhanced the oxidative stability and microbial quality of beef seekh kabab during storage. However, surface characterization methods such as scanning electron microscopy (SEM) and time-of-flight secondary ion mass spectrometry (TOF-SIMS) and the functional/bioactive properties of chitosan coating films could be considered in future studies.

## 4. Conclusions

This study concluded that chitosan coating can improve the physiochemical, microbial, and sensory quality of ready-to-cook beef seekh kabab during refrigerated storage. Among the treatments, 2% chitosan coating was identified as optimal, significantly reducing moisture loss, maintaining pH and color stability, and slowing down microbial growth, including TVCs, LAB, *Enterobacteriaceae*, and *S. aureus*. Furthermore, lipid oxidation, protein oxidation, and TVB-N levels were better controlled in the coated samples, supporting their role in preserving freshness. Sensory attributes were also retained more effectively in chitosan-coated samples compared with the control. The quality improvement was mainly due to the formation of a uniform chitosan layer acting as a selective oxygen and moisture barrier, along with the intrinsic antioxidant and antimicrobial properties of chitosan. Correlation analysis confirmed that chitosan coating allowed for 28 days of storage with minimal deterioration in key quality parameters. However, the coating had limited effect on preventing enzymatic lipid oxidation and proteolysis over time, as indicated by slight increases in TBARS and TVB-N. Overall, chitosan shows strong potential as a natural preservative for beef seekh kababs, offering promising industrial applications for shelf-life extension.

## Figures and Tables

**Figure 1 foods-14-03844-f001:**
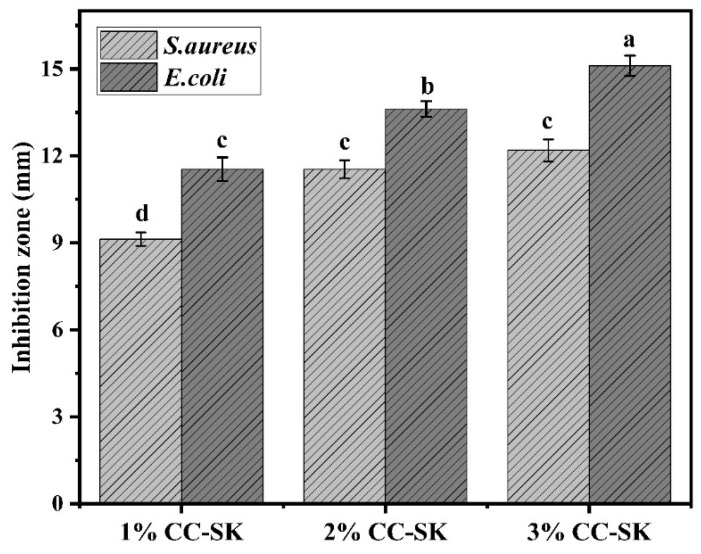
Different letters (a–d) denote statistically significant differences (*p* < 0.05) in the antimicrobial activity of the chitosan coating solutions against *S. aureus* and *E. coli*.

**Figure 2 foods-14-03844-f002:**
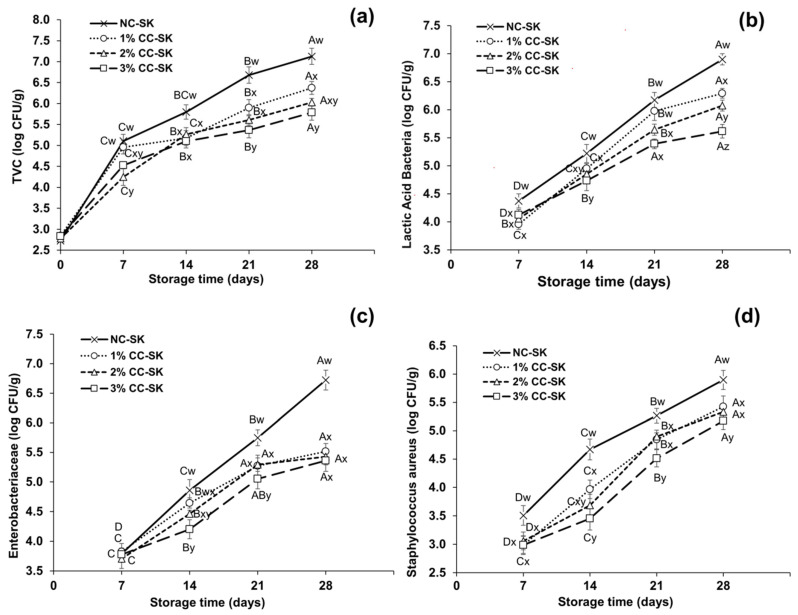
Effects of varying levels of chitosan coating on (**a**) total viable counts, (**b**) lactic acid bacteria, (**c**) *Enterobacteriaceae*, and (**d**) *Staphylococcus aureus* in beef seekh kabab during refrigerated storage at 4 °C. Different markers show the mean values of different treatments, while the bars indicate standard deviation at each storage time. A–D: Means with different letters indicate variations during storage time that differ significantly (*p* < 0.05), w–z: Means with different letters show that changes among treatments differ significantly (*p* < 0.05). Abbreviations: NC-SC: non-coated seekh kabab (control); CC-SK: chitosan-coated seekh kabab.

**Figure 3 foods-14-03844-f003:**
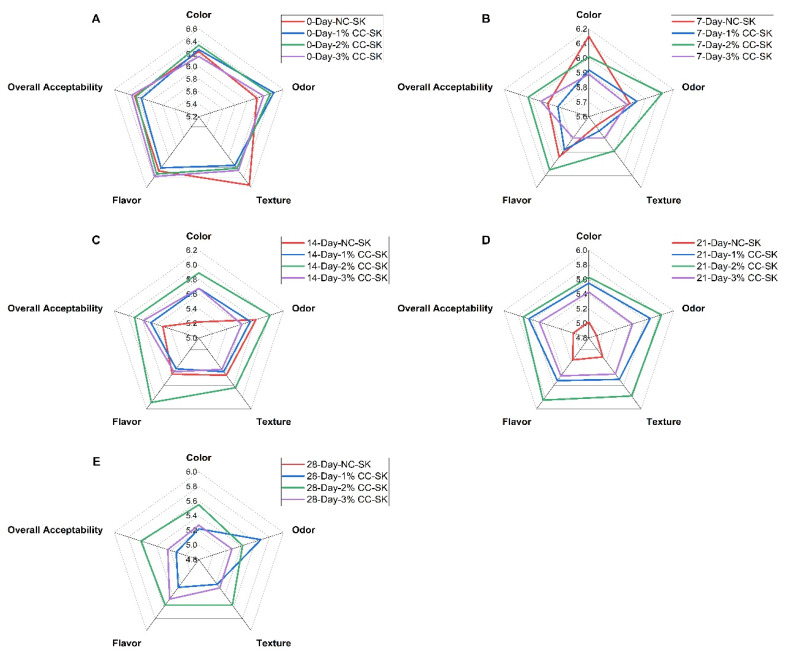
Effects of chitosan coating at varying concentrations on the sensory properties of ready-to-cook beef seekh kabab during refrigerated storage (4 °C) on day 0 (**A**), day 7 (**B**), day 14 (**C**), day 21 (**D**), and day 28 (**E**).

**Figure 4 foods-14-03844-f004:**
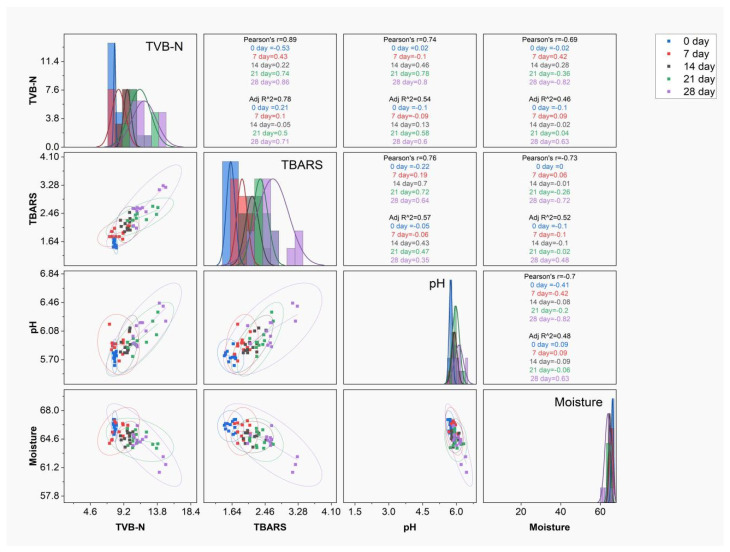
Scattered Pearson correlation matrix for TVB-N (mg/100 g), TBARS (mg MDA/kg), pH, and moisture in relation to the storage time of seekh kabab samples.

**Table 1 foods-14-03844-t001:** Ingredients for the preparation of seekh kabab.

Ingredients	For 1 kg of Seekh Kabab
Minced beef	800 g
2. Fat	200 g
3. Green chili	20 g
4. Onion	350 g
5. Ginger	12 g
6. Garlic	20 g
7. Salt	6 g
8. Cumin	2.5 g
9. Black pepper powder	2.7 g
10. Red chili powder	2.5 g
11. Coriander	5 g
12. Long pepper	1.5 g
13. Lime powder	0.5 g
14. Baking soda	3 g

**Table 2 foods-14-03844-t002:** Effects of varying levels of chitosan coating on the proximate composition of beef seekh kabab stored at 4 °C for 28 days.

Parameters	Treatments	Storage Time (Days)				
		0	7	14	21	28
Ash (%)	NC-SK	2.26 ± 0.14 ^A^	2.20 ± 0.09 ^ABw^	2.13 ± 0.16 ^ABw^	2.11 ± 0.11 ^ABw^	2.02 ± 0.11 ^Bw^
	1% CC-SK	2.20 ± 0.04 ^A^	2.06 ± 0.15 ^Awx^	2.05 ± 0.07 ^Awx^	1.96 ± 0.10 ^Awx^	2.09 ± 0.19 ^Aw^
	2% CC-SK	2.17 ± 0.11 ^A^	1.97 ± 0.11 ^Bx^	1.91 ± 0.16 ^BCx^	1.89 ± 0.14 ^Cx^	1.89 ± 0.23 ^Cwx^
	3% CC-SK	2.25 ± 0.09 ^A^	2.14 ± 0.15 ^ABwx^	1.98 ± 0.35 ^Bwx^	1.98 ± 0.50 ^Bwx^	1.94 ± 0.18 ^Bwx^
Moisture (%)	NC-SK	65.89 ± 0.65 ^A^	64.74 ± 0.59 ^ABx^	64.78 ± 0.41 ^ABw^	63.82 ± 0.39 ^By^	61.55 ± 0.74 ^Cx^
	1% CC-SK	66.68 ± 0.25 ^A^	66.51 ± 0.26 ^Aw^	65.46 ± 0.14 ^Bw^	65.42 ± 0.27 ^Bw^	64.54 ± 0.30 ^Cw^
	2% CC-SK	65.55 ± 0.24 ^A^	65.40 ± 0.46 ^Awx^	64.06 ± 0.42 ^Bx^	64.11 ± 0.26 ^Bwx^	64.25 ± 0.51 ^Bw^
	3% CC-SK	66.43 ± 0.46 ^A^	65.44 ± 1.01 ^Awx^	64.21 ± 0.34 ^Bx^	64.71 ± 0.85 ^Bwx^	64.84 ± 0.91 ^Bw^
Fat (%)	NC-SK	15.02 ± 0.58 ^A^	14.53 ± 0.33 ^ABw^	13.89 ± 0.27 ^B^	13.21 ± 0.3 ^Cx^	12.85 ± 0.36 ^Cx^
	1% CC-SK	15.17 ± 0.55 ^A^	14.51 ± 0.50 ^ABwx^	14.06 ± 0.31 ^B^	13.75 ± 0.62 ^BCw^	13.30 ± 0.14 ^Cw^
	2% CC-SK	14.96 ± 0.17 ^A^	14.69 ± 0.17 ^ABx^	14.19 ± 0.18 ^BC^	13.63 ± 0.18 ^CDw^	13.48 ± 0.18 ^Dw^
	3% CC-SK	15.22 ± 0.12 ^A^	14.79 ± 0.25 ^ABwx^	14.29 ± 0.45 ^B^	13.95 ± 0.38 ^Cw^	13.66 ± 0.10 ^Cw^
Protein (%)	NC-SK	18.22 ± 0.42 ^A^	17.39 ± 0.89 ^ABx^	16.70 ± 0.67 ^Bx^	15.82 ± 0.43 ^CDy^	15.44 ± 0.56 ^Dy^
	1% CC-SK	18.37 ± 0.36 ^A^	17.83 ± 0.48 ^ABwx^	16.91 ± 0.44 ^Bwx^	16.51 ± 0.47 ^BCx^	15.90 ± 0.17 ^Cxy^
	2% CC-SK	17.91 ± 1.25 ^A^	17.56 ± 0.85 ^ABx^	17.05 ± 0.67 ^Bw^	16.93 ± 0.20 ^Bw^	16.14 ± 0.60 ^Cwx^
	3% CC-SK	18.10 ± 0.37 ^A^	18.03 ± 0.25 ^Aw^	17.50 ± 0.24 ^ABw^	17.02 ± 0.46 ^Bw^	16.71 ± 0.37 ^BCw^

The data are presented as means ± standard deviations (*n* = 4). A–D: Means within the same row bearing different superscripts differ significantly (*p* < 0.05); w–y: Means within the same column bearing different superscripts differ significantly (*p* < 0.05). Abbreviations: NC-SC: non-coated seekh kabab (control); CC-SK: chitosan-coated seekh kabab.

**Table 3 foods-14-03844-t003:** Effects of varying levels of chitosan coating on pH and color values of beef seekh kabab stored at 4 °C for 28 days.

Parameters	Treatments	Storage Time (Days)				
		0	7	14	21	28
pH	NC-SK	5.80 ± 0.04 ^D^	6.03 ± 0.03 ^Cw^	6.09 ± 0.16 ^BCw^	6.18 ± 0.03 ^Bw^	6.37 ± 0.07 ^Aw^
	1% CC-SK	5.75 ± 0.03 ^C^	5.88 ± 0.03 ^Bx^	5.84 ± 0.04 ^Bxy^	5.89 ± 0.06 ^Bx^	6.16 ± 0.05 ^Ax^
	2% CC-SK	5.70 ± 0.04 ^B^	5.90 ± 0.05 ^Awx^	5.90 ± 0.06 ^Ax^	5.91 ± 0.04 ^Ax^	5.98 ± 0.07 ^Ay^
	3% CC-SK	5.72 ± 0.09 ^C^	5.77 ± 0.05 ^BCy^	5.81 ± 0.05 ^ABy^	5.86 ± 0.08 ^ABx^	5.92 ± 0.08 ^Ay^
L* (lightness)	NC-SK	43.08 ± 0.75 ^C^	44.02 ± 0.83 ^C^	44.44 ± 0.81 ^Cy^	46.71 ± 0.57 ^By^	48.26 ± 0.61 ^Ay^
	1% CC-SK	44.11 ± 1.10 ^CD^	45.02 ± 1.48 ^C^	45.76 ± 0.64 ^Cxy^	47.52 ± 1.85 ^Bwx^	49.12 ± 1.28 ^Awx^
	2% CC-SK	43.97 ± 0.60 ^C^	44.63 ± 0.39 ^C^	46.60 ± 1.12 ^Bwx^	49.29 ± 0.26 ^Aw^	50.11 ± 0.13 ^Awx^
	3% CC-SK	44.12 ± 1.03 ^C^	45.47 ± 1.11 ^C^	47.54 ± 0.34 ^BCw^	48.06 ± 0.85 ^Bwx^	51.43 ± 0.91 ^Aw^
a* (redness)	NC-SK	7.91 ± 0.66 ^A^	7.52 ± 0.46 ^A^	6.04 ± 0.35 ^B^	5.75 ± 0.62 ^Cx^	5.65 ± 0.66 ^Cx^
	1% CC-SK	8.01 ± 0.14 ^A^	7.86 ± 0.62 ^AB^	6.85 ± 0.51 ^B^	5.85 ± 0.50 ^Cx^	5.75 ± 0.55 ^Cwx^
	2% CC-SK	8.05 ± 0.27 ^A^	7.73 ± 0.74 ^AB^	6.14 ± 0.45 ^B^	6.19 ± 0.23 ^Bw^	6.11 ± 0.17 ^Bw^
	3% CC-SK	7.89 ± 0.10 ^A^	7.51 ± 0.34 ^AB^	6.08 ± 0.51 ^B^	6.17 ± 0.65 ^Cw^	5.89 ± 0.25 ^Cwx^
b* (yellowness)	NC-SK	18.22 ± 1.13 ^B^	18.75 ± 0.62 ^Bwx^	19.04 ± 1.21 ^AB^	21.18 ± 0.71 ^Aw^	21.78 ± 0.92 ^Aw^
	1% CC-SK	18.07 ± 1.14 ^B^	19.01 ± 0.37 ^Bw^	18.93 ± 0.73 ^B^	20.12 ± 0.98 ^Awx^	20.89 ± 0.73 ^Awx^
	2% CC-SK	18.43 ± 1.14 ^B^	18.54 ± 0.86 ^Bx^	18.87 ± 1.30 ^B^	19.68 ± 0.67 ^Ax^	20.14 ± 0.69 ^Ax^
	3% CC-SK	19.04 ± 0.76 ^AB^	19.24 ± 0.22 ^ABw^	18.98 ± 0.51 ^AB^	20.13 ± 0.65 ^Awx^	20.08 ± 0.25 ^Ax^

The data are presented as means ± standard deviations (*n* = 4). A–D: Means within the same row bearing different superscripts differ significantly (*p* < 0.05); w–y: Means within the same column bearing different superscripts differ significantly (*p* < 0.05). Abbreviations: NC-SC: non-coated seekh kabab (control); CC-SK: chitosan-coated seekh kabab.

**Table 4 foods-14-03844-t004:** Effects of varying levels of chitosan coating on the TBARS, carbonyl content, and TVB-N of beef seekh kabab stored at 4 °C for 28 days.

Parameters	Treatments	Storage Time (Days)				
		0	7	14	21	28
TBARS (mg MDA/kg)	NC-SK	1.57 ± 0.11 ^D^	1.91 ± 0.08 ^Cw^	2.32 ± 0.14 ^Bw^	2.57 ± 0.12 ^Bw^	3.20 ± 0.04 ^Aw^
	1% CC-SK	1.62 ± 0.12 ^D^	1.96 ± 0.17 ^Cw^	2.08 ± 0.15 ^Cx^	2.33 ± 0.08 ^Bx^	2.57 ± 0.08 ^Axy^
	2% CC-SK	1.59 ± 0.12 ^D^	1.86 ± 0.08 ^Cx^	2.06 ± 0.13 ^Cx^	2.26 ± 0.18 ^Ax^	2.21 ± 0.25 ^Ay^
	3% CC-SK	1.64 ± 0.04 ^D^	1.80 ± 0.11 ^Cx^	2.07 ± 0.17 ^Bx^	2.19 ± 0.10 ^Bx^	2.61 ± 0.19 ^Ax^
Carbonyl content (nmol/mg protein)	NC-SK	2.23 ± 0.09 ^E^	2.81 ± 0.18 ^Dw^	3.22 ± 0.16 ^Cw^	3.65 ± 0.16 ^Bw^	3.96 ± 0.13 ^Aw^
	1% CC-SK	2.26 ± 0.17 ^D^	2.73 ± 0.13 ^Cwx^	2.98 ± 0.09 ^BCx^	3.11 ± 0.11 ^Bx^	3.62 ± 0.11 ^Ax^
	2% CC-SK	2.30 ± 0.12 ^D^	2.61 ± 0.08 ^Cx^	2.81 ± 0.07 ^BCy^	3.06 ± 0.12 ^Ax^	3.42 ± 0.15 ^Ay^
	3% CC-SK	2.28 ± 0.04 ^D^	2.66 ± 0.13 ^BCx^	2.87 ± 0.12 ^By^	3.16 ± 0.15 ^Axy^	3.34 ± 0.14 ^Ay^
TVBN (mg/100 g)	NC-SK	7.96 ± 0.20 ^D^	8.73 ± 0.03 ^Dw^	10.01 ± 0.04 ^Cw^	13.58 ± 0.13 ^Bw^	14.55 ± 0.12 ^Aw^
	1% CC-SK	7.86 ± 0.09 ^C^	8.64 ± 0.18 ^Cw^	9.96 ± 0.43 ^Bwx^	11.35 ± 0.43 ^Ax^	11.84 ± 0.46 ^Ax^
	2% CC-SK	7.99 ± 0.30 ^C^	8.12 ± 0.41 ^BCx^	9.43 ± 0.27 ^Bwx^	10.42 ± 0.23 ^Ay^	10.83 ± 0.35 ^Ay^
	3% CC-SK	8.00 ± 0.13 ^C^	8.32 ± 0.29 ^BCx^	9.29 ± 0.53 ^By^	10.21 ± 0.44 ^Ay^	10.73 ± 0.26 ^Ay^

The data are presented as means ± standard deviations (*n* = 4). A–D: Means within the same row bearing different superscripts differ significantly (*p* < 0.05); w–y: Means within the same column bearing different superscripts differ significantly (*p* < 0.05). Abbreviations: NC-SC: non-coated seekh kabab (control); CC-SK: chitosan-coated seekh kabab.

## Data Availability

Data are available from the corresponding author upon request.

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
