# Peer review of "Impacts of Chitosan Coating on Shelf Life and Quality of Ready-to-Cook Beef Seekh Kabab During Refrigeration Storage"

_foods, 2025, doi:10.3390/foods14223844_

Round 1

Reviewer 1 Report

Comments and Suggestions for Authors

The manuscript presents data regarding the use of chitosan to preserve beef seekh kabab. It is an original topic and relevant to the field. Chitosan is widely used as coating for different foodstuff, however, information regarding chitosan coating on beef is absent. The manuscript is well written, scientifically sound and design was properly accomplished. The research addressed different chitosan concentrations (1, 2 and 3%) and evaluated several beef characteristics in those different concentrations, and this information is important for the food industry. Other published material did not evaluate different chitosan concentrations in beef seekh kabab. Material and methods require minor alterations, as indicated at the suggestions for authors section. An important information is required: if the research was submitted to the ethics committee approval prior to its accomplishment. is crucial to analyze the further acceptance or not, because the research used panelists to assess sensorial attributes. The authors should indicate the Proof of Application for Ethical Review number.

Keywords: beef seekh kabab; chitosan coating; quality; refrigeration must be altered, because these words are already used in the manuscripts title. This is required for indexing purposes.

Introduction: it is well written, indicating the importance of the research, with proper background. One suggestion: as authors stated that beef seekh kabab is increasingly its consumption, data to consolidade this phrase should be inserted, such as production, consumption  in the last 5 years, and main countries that are using it.

 Materials and Methods:

  1. line 98: indicate if moisture is wet or dry basis.
  2. line 115: indicate brand/model of the stirrer and the velocity used
  3. lines 115-117: insert explanation why the authors used "plasticizer (consisting of 50% polyethylene glycol and 50% glycerol) was added with 1% of acetic
    acid into the chitosan solution"
  4. line 238: the panelists signed an informed term of consent? the research was submitted to the Research Ethics Committee? please provide the process number

Author Response

Reviewer 1

The manuscript presents data regarding the use of chitosan to preserve beef seekh kabab. It is an original topic and relevant to the field. Chitosan is widely used as coating for different foodstuff, however, information regarding chitosan coating on beef is absent. The manuscript is well written, scientifically sound and design was properly accomplished. The research addressed different chitosan concentrations (1, 2 and 3%) and evaluated several beef characteristics in those different concentrations, and this information is important for the food industry. Other published material did not evaluate different chitosan concentrations in beef seekh kabab. Material and methods require minor alterations, as indicated at the suggestions for authors section. An important information is required: if the research was submitted to the ethics committee approval prior to its accomplishment. is crucial to analyze the further acceptance or not, because the research used panelists to assess sensorial attributes. The authors should indicate the Proof of Application for Ethical Review number.

Response: We are very grateful for taking the time and give us valuable comments and suggestions to further improve and increase the quality of the manuscript. The informed consent was obtained from all participants prior to tasting the samples, and approval was secured from the Research Ethics Committee. This information has been included in the revised manuscript in line 249-252.

  1. Keywords: beef seekh kabab; chitosan coating; quality; refrigeration must be altered, because these words are already used in the manuscripts title. This is required for indexing purposes.

Response: Thanks for your comment! The keywords have been revised according to your suggestion. As follows in Line 34.

“Keywords: beef; ready-to-cook; coating; lipid oxidation; antimicrobial activity; storage”

  1. Introduction: it is well written, indicating the importance of the research, with proper background. One suggestion: as authors stated that beef seekh kabab is increasingly its consumption, data to consolidade this phrase should be inserted, such as production, consumption in the last 5 years, and main countries that are using it.

Response: We really appreciate for raising this comment. It helps us to improve the background data on Seekah kebab. We have made the following modifications in the introduction section Line 37—45.

“According to the latest research, the global kebab shop market size reached USD 21.7 billion in 2024, reflecting a robust foodservice sector worldwide. The market is experiencing a steady expansion, registering a compound annual growth rate (CAGR) of 6.2% from 2025 to 2033. https://dataintelo.com/report/kebab-shop-market”

  1. Materials and Methods: line 98: indicate if moisture is wet or dry basis.

Response: Thank you for your comment!  The moisture content of the chitosan powder is expressed on a wet basis. This information has been included in the revised manuscript in line 97.

“Low molecular weight chitosan powder (Food grade, CAS No. 70694-72-3, molecular weight 50–190 kDa, Batch No, 230223) sourced from crustacean shells with a deacetylation degree of 95%, purity of 87.7% and moisture below 10% (dry basis) was purchased from Qingdao Honghai Bio-Tec Co., Ltd. Qingdao, China.”

  1. line 115: indicate brand/model of the stirrer and the velocity used

Response: Thank you for the valuable suggestion. The model and operating condition for the stirrer have been added accordingly in the revised manuscript in line 114-115.

“In detail, different levels of chitosan at 0%, 1%, 2%, and 3% were dissolved in distilled water using a stirrer (INTLLAB Magnetic Stirrer Mixer 3000 rpm, Guangdong, China) for 10 min at 60 ℃.”

  1. lines 115-117: insert explanation why the authors used "plasticizer (consisting of 50% polyethylene glycol and 50% glycerol) was added with 1% of acetic acid into the chitosan solution"

Response: Thanks for your comment! The use of polyethylene glycol mixture with acetic acid was intended to optimize the mechanical and functional properties of the chitosan coating whereas, 1% acetic acid was used to dissolve the chitosan and ensure uniform dispersion of the plasticizer. This information has also been added in the revised manuscript in line 117-119 as follows.

“The use of plasticizer was intended to optimize the mechanical and functional properties of the chitosan coating whereas, 1% acetic acid was used to dissolve the chitosan and ensure uniform dispersion of the plasticizer.”

  1. line 238: the panelists signed an informed term of consent? the research was submitted to the Research Ethics Committee? please provide the process number.

Response: Thank you for your comment! Yes, informed consent was obtained from all participants prior to tasting the samples, and approval was secured from the Research Ethics Committee. This information has been included in the revised manuscript in line 249-252.

“Sensory analysis was conducted in accordance with the Declaration of Helsinki, and approved by the Ethics Committee of the Institute of Food Science and Technology, Chinese Academy of Agricultural Sciences (CAAS-IFST & DLS-2025/267 on 5 April, 2025).”

Reviewer 2 Report

Comments and Suggestions for Authors

Dear authors,

after review I list my suggestions below:

59-75 There is wide description of chitosan properties and mechanism of action which should be included in discussion section. The introduction section should Focus on explaining the problem, highlighting the importance of the study and providing short literature review.

97-99 There is lack of chitosan molecular weight which is important parameter, especially regarding to antibacterial properties. The term ‘low molecular weight’ is imprecise.

Additionally, there should be information about the chitosan origin (shrimp, fungi, etc..)

145 ‘Laminar airflow’ is imprecise. Please provide the flow parameters.

146-147 There is no information about zip locker bags used for kabab storage. These information should be provided, especially parameters describing gas (oxygen) exchange between inner and outter bag environment as it can affect antioxidation and/or anti microbial properties.

155 Anti-pathogenicity has ambiguous meaning. I suggest ‘anti microbal activity’.

I highly recommend characteristics of chitosan coating using scanning electron microscopy (SEM) and/or secondary ion mass spectrometry with time of flight detection (TOF-SIMS). This approach could help better understand the structure and behaviour of chitosan coating as well as how it interacts with the kabab surface.

The conclusion section lacks syntetic results e.g. which chitosan concentration was found to be optimal for this application or which mechanims of chitosan were the most important.

There is lack of discussion about limitations of the study e.g. influence of zip-lock bags or number of sensory assessors.

Author Response

Reviewer 2

Dear authors, after review I list my suggestions below:

  1. 59-75 There is wide description of chitosan properties and mechanism of action which should be included in discussion section. The introduction section should focus on explaining the problem, highlighting the importance of the study and providing short literature review.

Response: Thanks for your comment! The introduction part has been revised and some of the content has been adjusted in the discussion part in lipid oxidation in line 434-438 and antimicrobial activity 292-296.

“Line: 290-294 Chitosan functions as an antibacterial agent through electrostatic interactions between its positively charged amino groups and the negatively charged microbial cell mem-branes, increasing membrane permeability, causing leakage into intracellular contents, and ultimately resulting in cell death [27, 12]”

“Line 432-437 The antioxidant activity of chitosan is mainly due to the presence of free amino (-NH2) and hydroxyl (-OH) which act as hydrogen donors to neutralize lipid radicals and interrupt oxidative chain reactions. Moreover, chitosan can chelate pro-oxidant metal ions such as Fe2+ and Cu2+, thereby inhibiting the Fenton reaction that generates reactive oxygen species [47]”

  1. 97-99 There is lack of chitosan molecular weight which is important parameter, especially regarding to antibacterial properties. The term ‘low molecular weight’ is imprecise.

Response: Thanks for your comment! The chitosan used in our study had a molecular weight of 50–190 kDa with a degree of deacetylation of 95%. This information has been added in the revised manuscript in line 96-97.

“Low molecular weight chitosan powder (Food grade, CAS No. 70694-72-3, molecular weight 50–190 kDa, Batch No, 230223) sourced from crustacean shells with a deacetylation degree of 95%, …”

  1. Additionally, there should be information about the chitosan origin (shrimp, fungi, etc..)

Response: Thanks for your comment! The information has been added in the revised manuscript in line 96.

“Low molecular weight chitosan powder (Food grade, CAS No. 70694-72-3, molecular weight 50–190 kDa, Batch No, 230223) sourced from crustacean shells with a deacetylation degree of 95%, …”

  1. 145 ‘Laminar airflow’ is imprecise. Please provide the flow parameters.

Response: Thanks for your comment! The information has been added in the manuscript in lines 147-149.

“The precooked seekh kabab samples were dipped in a chitosan solution (1%, 2%, and 3%) for 2 minutes to ensure complete coating. The samples were then drained, and allowed to fully dry in a laminar airflow (HH 48, Holten LaminAir, Thermo Fisher Scientific, Bath, UK) at an air velocity of 0.35-0.45ms-1 for 30 min.”

  1. 146-147 There is no information about zip locker bags used for kabab storage. These information should be provided, especially parameters describing gas (oxygen) exchange between inner and outter bag environment as it can affect antioxidation and/or antimicrobial properties.

Response: Thanks for your comment! The details on the zip bags used are included accordingly in line 150-151.  

“Following this, the samples were properly labeled and then packaged in commercial low-density polyethylene (LDPE) plastic zip locker bags (thickness 40-50 µm, O2 permeability, 80mL O2/m2/day⁻¹) and stored at 4 °C for intervals of 0, 7, 14, 21, and 28 days.”

  1. 155 Anti-pathogenicity has ambiguous meaning. I suggest ‘anti microbal activity’.

Response: Thanks for your comment! The suggested method name has been revised accordingly in line 160.

“2.5.    Antimicrobial activity of chitosan coatings”

  1. I highly recommend characteristics of chitosan coating using scanning electron microscopy (SEM) and/or secondary ion mass spectrometry with time of flight detection (TOF-SIMS). This approach could help better understand the structure and behavior of chitosan coating as well as how it interacts with the kabab surface.

Response: We sincerely thank the reviewer for this valuable suggestion. Although SEM or TOF-SIMS analyses would indeed provide microstructural information. However, due to resource constraints these experiments could not be performed at this stage. This limitation has been acknowledged, and we have suggested these analyses as part of future research. The information has been added in line 613-617.

“The present study showed that chitosan coating successfully enhanced the oxidative stability and microbial quality of beef seekh kabab during storage. However, surface characterization methods such as scanning electron microscopy (SEM) and time of flight secondary ion mass spectrometry (TOF-SIMS) and functional/bioactive properties of chitosan coating films could be considered for future studies.”

  1. The conclusion section lacks synthetic results e.g. which chitosan concentration was found to be optimal for this application or which mechanims of chitosan were the most important.

Response: Thanks for your comment! The conclusion part has been revised according to your suggestion in line 621-623.

“Among the treatments, the 2 % chitosan coating was identified as optimal, significantly reducing moisture loss, maintaining pH and color stability, and slowing microbial growth, including TVC, LAB, Enterobacteriaceae, and S. aureus.”

  1. There is lack of discussion about limitations of the study e.g. influence of zip-lock bags or number of sensory assessors.

Response: We appreciate the reviewer’s suggestion. We have added a discussion acknowledging that the use of polyethylene freezer bags may have influenced volatile compound retention, in line 570-571.

“It was reported that the use of polyethylene freezer bags may have influenced on volatile compound retention, moisture, and sensory attributes [59].”

  • Torri, L., & Piochi, M. (2016). Sensory methods and electronic nose as innovative tools for the evaluation of the aroma transfer properties of food plastic bags. Food Res. Int., 85, 235–243. https://doi.org/10.1016/j.foodres.2016.03.00

Reviewer 3 Report

Comments and Suggestions for Authors

foods-3967059

The study demonstrates that 2–3% chitosan coatings effectively extend shelf life of beef seekh kababs by reducing microbial growth and oxidation while maintaining quality parameters. While the practical findings are valuable for food preservation, the novelty is limited given extensive prior work on chitosan coatings in meat products. The manuscript needs clearer articulation of its unique contribution and specific advances beyond existing research. Minor revision recommended.

Line 86 How do these concentration levels compare with those commonly used in chitosan coatings for meat or poultry products?

Line 102 What muscle part of the carcass was used for seekh kabab preparation? specific cut or muscle type used e.g., semimembranosus or longissimus dorsi.

Line 155 Was acetic acid alone tested as a control to distinguish chitosan effect? since it may itself inhibit bacteria.

Line 283 How does the current antimicrobial performance compare quantitatively (inhibition zone diameter, CFU reduction) with literature using similar concentrations and Gram types?

Line 387 How do the reported L*, a*, b* values compare numerically to other studies on coated meat systems?

Line  415 The mechanism “chelating transition metals” should be supported by chemical evidence, did the authors analyze Fe²⁺/Fe³⁺ or radical scavenging activity of CC-SK film?

Line  438  What explains the similar effectiveness of 2% and 3% chitosan? Is there a saturation or diffusion limitation in the coating?

Line 531 What is the acceptability threshold (< 4 points), is this based on a specific standard?

Line  551 How was the correlation computed, across all storage times or at individual time points? If at 28 days, how does that represent temporal correlation?

Author Response

Reviewer 3

The study demonstrates that 2–3% chitosan coatings effectively extend shelf life of beef seekh kababs by reducing microbial growth and oxidation while maintaining quality parameters. While the practical findings are valuable for food preservation, the novelty is limited given extensive prior work on chitosan coatings in meat products. The manuscript needs clearer articulation of its unique contribution and specific advances beyond existing research. Minor revision recommended.

Response: We are very grateful for taking the time and review the manuscript, which helped us to shape and clarify most of the indicated comments.

  1. Line 86 How do these concentration levels compare with those commonly used in chitosan coatings for meat or poultry products?

Response: Thank you for raising this question. The concentrations of chitosan used in this study (1–3%) are in line with commonly applied ranges for meat and poultry coatings, as previous studies have shown that coatings within this range effectively reduce microbial growth, delay oxidation, and preserve sensory attributes (Dong et al., 2020; Zhang et al., 2018; Mithushan et al., 2024). To further clarify this point the following information has been added in the introduction section in lines 85-92.

“The selection of concentrations was based on previous studies [17,18] and preliminary tests. It was observed that chitosan concentrations above 3% led to undesirable changes in the surface appearance and altered taste (slightly bitter) of the meat, potentially affecting consumer acceptability (data not shown). Conversely, concentrations below 1% coating exhibited limited antimicrobial efficacy. Therefore, chitosan concentrations ranging from 1% to 3% were selected as the treatment levels in this study to determine the optimal concentration for enhancing the microbial stability, quality, and shelf life of the coated meat samples.”

In addition, the comparison between different samples were made in the result and discussion section. To mention some of the cited references in Line 289-293 in section 3.1; in Line 313-316 in section 3.2 with respect to the retention potential of moisture content, the comparison in the reduction of TBRS is also highlighted in line 426-434

“These findings are aligned with [19] who applied carboxymethyl chitosan coating solution with the concentrations 2% and 4% to mango fruit and reported significant shelf-life extension, supporting the suitability of these coating levels against Gram-positive (S. aureus) and Gram negative (E. coli) bacterial strains for food preservation applications.”

“In other related studies, [3] noted similar results in the loss of moisture content in the chevon kabab meat product during storage. The higher moisture content was noted in CC-SK samples during storage, which might be due to the water retention property of chitosan.”

“de Lima et al., (2024) also highlighted that chitosan slow down the lipid oxidation to 1.18 mg/kg while the TBARS values in uncoated meat samples reached to 1.66 mg/kg, exceeding sensory thresholds [45].”

  • Zhang, H., He, P., Kang, H., & Li, X. (2018). Antioxidant and antimicrobial effects of edible coating based on chitosan and bamboo vinegar in ready-to-cook pork chops. LWT. 93, 470–476. https://doi.org/10.1016/j.lwt.2018.03.061
  • Dong, C., Wang, B., Li, F., Zhong, Q., Xia, X., & Kong, B. (2020). Effects of edible chitosan coating on Harbin red sausage storage stability at room temperature. Meat Science, 159, 107919. https://doi.org/10.1016/j.meatsci.2019.107919
  • Mithushan, P., Pagthinathan, M., Ravikumar, S., & Vahaful Nisath, M. F. (2024). Effect of coating chemically derived chitosan from shrimp shell wastes on physical, microbiological and sensory characteristics of chicken sausages. Tropical Agricultural Research and Extension, 27, 183–191. https://doi.org/10.4038/tare.v27i3.5589.

  1. Line 102 What muscle part of the carcass was used for seekh kabab preparation? specific cut or muscle type used e.g., semimembranosus or longissimus dorsi.

Response: Thanks for your comment! The specific cut has been added in the line 102-103.  

“The beef meat (longissimus thoracis et lumborum (LTL)) and fat were collected from local market and placed in an icebox after packing in a zip locker bag, and shifted to the lab within 2 h of collection.”

  1. Line 155 Was acetic acid alone tested as a control to distinguish chitosan effect? since it may itself inhibit bacteria.

Response: Thank you for your comment! Even though acetic acid has antimicrobial activity at higher concentrations (≥2 %) (El Asuoty et al., 2023). However, in the present study, the 1 % solution served only to dissolve chitosan and was not tested as an independent control. Therefore, the improvements in microbial stability and oxidative quality witnessed in coated samples are mainly attributed to chitosan and its known mechanisms, including its polycationic nature, film-forming barrier effect, and metal-ion chelation (Rabea et al., 2003).”

  • El Asuoty, M. S., El Tedawy, F. A., & Abou‑Arab, N. M. (2023). Effect of thyme oil and acetic acid on the quality and shelf life of fresh meat. Journal of Advanced Veterinary Research, 13(6), 1079‑1083.
  • Rabea, E. I., Badawy, M. E. I., Stevens, C. V., Smagghe, G., & Steurbaut, W. (2003). Chitosan as antimicrobial agent: applications and mode of action. Biomacromolecules, 4(6), 1457–1465.

  1. Line 283 How does the current antimicrobial performance compare quantitatively (inhibition zone diameter, CFU reduction) with literature using similar concentrations and Gram types?

Response: Thank you for the comment! The chitosan concentrations used in the present study were similar to previously reported work by Ashraf et al. (2025), who observed comparable antimicrobial activity at similar concentration levels. In the present study, inhibition zone diameters for Staphylococcus aureus ranged from 9.13 ± 0.23 to 12.34 ± 0.25 mm, while those for Escherichia coli ranged from 11.67 ± 0.31 to 15.23 ± 0.27 mm, respectively indicating effective inhibition against both Gram-positive and Gram-negative bacteria. Furthermore, Ashraf et al. (2025) applied carboxymethyl chitosan coating at equivalent concentrations to mango fruit and reported significant shelf-life extension, supporting the suitability of these concentration levels against Gram positive and Gram negative bacterial strains for food preservation applications. The information has been added in Line 289-296.

  • Ashraf, J., Ismail, N., Tufail, T., Zhang, J., Awais, M., Zhang, Q., Ahmed, Z., Qi, Y., Liu, S., & Xu, B. (2025). Fabrication of novel pullulan/carboxymethyl chitosan-based edible film incorporated with ultrasonically equipped aqueous zein/turmeric essential oil nanoemulsion for effective preservation of mango fruits. International Journal of Biological Macromolecules, 294, 139330. https://doi.org/10.1016/j.ijbiomac.2024.139330

  1. Line 387 How do the reported L*, a*, b* values compare numerically to other studies on coated meat systems?

Response: We thank the reviewer for this valuable comment. The oxidation of lipids and heme-proteins in meat occurs simultaneously, prompting color alterations (L*, a*, b*). In the present study, the initial instrumental color values were L* values ranging from 43 to 44, a* from 7.89 to 8.05, and b* from 18.07 to 19.05 L*. These results are in line with previous studies, which reported 0-day L* values ranging from 43.19 to 44.09; similarly, the a* values were in the range of 9.33 to 10.61, and b* values from 21.35 to 22.96 in the chitosan-coated chevon kabab samples. Similarly, Farina et al. (2022) recorded that the L* values for un coated sample was 43.34 in beef patties with a slight increase after chitosan coating. Chitosan coating improves the color properties of meat products by reducing oxidative pigment degradation (Pabast et al., 2018). These findings suggested that applying chitosan coating could effectively reduce pigment oxidation, moisture loss, preserving the natural surface and redness of coated meat products. This information has been added in lines 354-361.

  • Farina, P., Ascrizzi, R., Bedini, S., Castagna, A., Flamini, G., et al. (2022). Chitosan and essential oils combined for beef meat protection against oviposition, water loss, lipid peroxidation, and colour changes. Foods, 11(24), 3994. https://doi.org/10.3390/foods11243994
  • Mishra, V., Tarafdar, A., Talukder, S., et al. (2023). Enhancing the shelf life of chevon Seekh Kabab using chitosan edible film and Cinnamomum zeylanicum essential oil. Journal of Food Science and Technology, 60, 1814–1825. https://doi.org/10.1007/s13197-023-05723-1
  • Pabast, M., Shariatifar, N., Beikzadeh, S., & Jahed, G. (2018). Effect of chitosan coatings and nano-encapsulated essential oils on quality characteristics of lamb meat. Food Control, 92, 37–45.

  1. Line 415 The mechanism “chelating transition metals” should be supported by chemical evidence, did the authors analyze Fe²⁺/Fe³⁺ or radical scavenging activity of CC-SK film?

Response: Thanks for your comment! In the present study, Fe²⁺/Fe³⁺ levels and radical scavenging activity of the chitosan-coated seekh kababs were not directly analyzed; therefore, the metal-chelating mechanism was proposed based on previously established evidence. We also added a clarifying statement in the revised discussion and acknowledged that future studies should quantify Fe²⁺/Fe³⁺ and assess radical scavenging activity to confirm the proposed mechanism for the chitosan coating used in this study.  This information has been added in the revised manuscript lines 434-440.

“The antioxidant activity of chitosan is mainly due to the presence of free amino (-NH2) and hydroxyl (-OH), which act as hydrogen donors to neutralize lipid radicals and interrupt oxidative chain reactions. Moreover, chitosan can chelate pro-oxidant metal ions such as Fe2+ and Cu2+, thereby inhibiting the Fenton reaction that generates reactive oxygen species [47]. By chelating transition metal ions or by interacting with malondialdehyde through its main amino group, chitosan can prevent lipid oxidation in meat products [33].”    

  • de Lima, A. F., Leite, R. H. d. L., Pereira, M. W. F., Silva, M. R. L., de Araújo, T. L. A. C., de Lima Júnior, D. M., Gomes, M. d. N. B., & Lima, P. d. O. (2024). Chitosan coating with rosemary extract increases shelf life and reduces water losses from beef. Foods, 13(9), 1353. https://doi.org/10.3390/foods13091353.
  • Zhang, X., Ismail, B. B., Cheng, H., Jin, T. Z., Qian, M., Arabi, S. A., Liu, D., & Guo, M. (2021). Emerging chitosan–essential oil films and coatings for food preservation: A review of advances and applications. Carbohydrate Polymers, 273, 118616. https://doi.org/10.1016/j.carbpol.2021.118616

  1. Line 438 What explains the similar effectiveness of 2% and 3% chitosan? Is there a saturation or diffusion limitation in the coating?

Response: Thanks for your comment! The similar efficiency witnessed among 2% and 3% chitosan levels may be attributed to a diffusion limitation and surface saturation effect. At higher levels, chitosan tends to increase solution viscosity and form a thicker coating layer, which may impede uniform diffusion and interaction with the meat surface, leading to no further improvement in antioxidant or antimicrobial efficiency. The following information has been added in lines 463-467.

“The effectiveness of 2% and 3% chitosan concentrations against protein oxidation might be due to the solution viscosity, which forms a thicker coating layer. Similarly, it was reported that increasing chitosan beyond an optimal level did not significantly influence on the preservation because of limited oxygen permeability and reduced bioactive compound migration.”

Similar trends have been reported in previous studies, where increasing chitosan beyond an optimal level (typically 1–2%) did not significantly enhance preservation effects due to limited oxygen permeability and reduced bioactive compound migration (Siripatrawan & Harte, 2010; da Silva et al., 2022; de Lima et al., 2024). 

  • Siripatrawan, U., & Harte, B. R. (2010). Physical properties and antioxidant activity of an active film from chitosan incorporated with green tea extract. Food Hydrocolloids, 24(8), 770–775.
  • da Silva, C. M., de Oliveira, R. A., Pagno, C. H., & de Lima, V. A. (2022). Chitosan coatings for meat preservation: Mechanisms, recent advances and future trends. Food Chemistry, 373, 131508.
  • de Lima, A. F., et al. (2024). Chitosan coating with rosemary extract increases shelf life and reduces water losses from beef. Foods, 13(9), 1353.

  1. Line 531 What is the acceptability threshold (< 4 points), is this based on a specific standard?

Response: We thank the reviewer for this valuable comment. In the current study, we have used 7-point numerical scale, where 7 corresponded to a most liked sample and 0 corresponding to a least liked sample. This part has been updated in the revised manuscript lines 558-559.  

“Whereas, the score of 4 was taken as the lower limit of acceptability (neither like nor dis-like) [25].”

  • Kanatt, S. R., Rao, M. S., Chawla, S. P., & Sharma, A. (2013). Effects of chitosan coating on shelf-life of ready-to-cook meat products during chilled storage. LWT - Food Science and Technology, 53(1), 321–326. https://doi.org/10.1016/j.lwt.2013.01.019

  1. Line 551 How was the correlation computed, across all storage times or at individual time points? If at 28 days, how does that represent temporal correlation?

Response: Thanks for your comment! The correlation was done using Pearson correlation by considering different time points and key quality parameters. The bar graph represents the correlation at different time points within same square to compare between two variables, while the lines represent peak correlation index. The Pearson relationship between different measurement points are represented with eclipse and measurement points at the bottom.

Round 2

Reviewer 2 Report

Comments and Suggestions for Authors

Thank you for clarification and paper improvement.